# Mixtape: Breaking the Softmax Bottleneck Efficiently

**Zhilin Yang[1], Thang Luong[2], Ruslan Salakhutdinov[1], Quoc Le[2]**
[1]Carnegie Mellon University, [2]Google Brain
{zhiliny,rsalakhu}@cs.cmu.edu, {thangluong,qvl}@google.com

## Abstract

The softmax bottleneck has been shown to limit the expressiveness of neural language models. Mixture of Softmaxes (MoS) is an effective approach to address such a theoretical limitation, but are expensive compared to softmax in terms of both memory and time. We propose Mixtape, an output layer that breaks the softmax bottleneck more efficiently with three novel techniques—logit space vector gating, sigmoid tree decomposition, and gate sharing. On four benchmarks including language modeling and machine translation, the Mixtape layer substantially improves the efficiency over the MoS layer by 3.5x to 10.5x while obtaining similar performance. A network equipped with Mixtape is only 20% to 34% slower than a softmax-based network with 10-30K vocabulary sizes, and outperforms softmax in perplexity and translation quality.

## 1   Introduction

Softmax has been a standard output layer for a wide variety of neural networks, including the majority of neural language models [5, 2, 3, 8, 11]. However, as pointed out by [19], softmax is a fundamental limitation of the expressiveness of neural language models, because it constrains the output representations to be low-rank, which might not be sufficient for modeling the complexity of natural language. Such a limitation is called the softmax bottleneck. To break the softmax bottleneck, [19] proposed Mixture of Softmaxes (MoS) that introduces discrete latent variables into the output layer so that the log probability matrix is high-rank because of the log-sum-exp nonlinear transformation. However, MoS is expensive compared to softmax in terms of both memory and time, which makes it less practically useful when computational budgets are limited.

To reduce the computational cost of MoS, we propose a novel output layer Mixtape to break the softmax bottleneck efficiently. Mixtape can be plugged into any existing networks as an additional layer before the cross entropy loss. Instead of employing a scalar mixture in the probability space as in MoS, Mixtape applies a vector gating mechanism in the logit space to avoid using multiple expensive softmaxes. In addition, Mixtape uses two more novel techniques to further reduce the computational cost. First, the vector gating mechanism is expensive because we need to compute a softmax gate for each word in the vocabulary. We propose *sigmoid tree decomposition* that decomposes a softmax probability gating distribution into a depth-2 binary tree structure, where each branch carries a portion of the probability mass determined by a sigmoid function. Sigmoid tree decomposition is much more efficient because it avoids the reduction and division operations in softmax. The other technique *gate sharing* is to share the gate values for all infrequent words, resulting in partially high-rank representations. This technique saves a considerable amount of memory and computation without affecting the performance because the gate values of infrequent words are usually hard to accurately estimate even without sharing the gates.

With all the above techniques combined, Mixtape substantially improves the efficiency of MoS while obtaining comparable or even better performances on four benchmarks, including language modeling and machine translation. With normal vocabulary sizes (e.g., 10K-30K), the Mixtape layer is 1.6x to 11.5x fater than the MoS layer given the same batch size, and is 3.5x to 10.5x faster given the

same memory budget. With normal vocabulary sizes, a Mixtape-based network is only 5% to 18% slower than a softmax-based network given the same batch size, and is only 20% to 34% slower given the same memory budget. With a large vocabulary of 100K tokens, a Mixtape-based network is still only 60% slower than a softmax-based network. Both Mixtape and MoS outperform softmax in perplexity and translation quality. Interestingly, these benchmarks have varied vocabulary sizes ranging from 10K to 100K and different input representations including words and BPE subwords, which demonstrates that Mixtape is effective and robust with a variety of inputs.

## 2   Softmax Bottleneck

In the following, we will introduce the notations and review the softmax bottleneck problem pointed out by [19].

Consider a general setting of language modeling and text generation, where given the context $C$ we want to estimate the conditional distribution of the next token $P^*(X|C)$. Here we use $P^*$ to denote the true data distribution. The context $C$ denotes the tokens that have occurred so far. For example, given a corpus $(X_1, X_2, \cdots, X_T)$, for each time step $t$, we aim to estimate the probability $P^*(X_t|C = X_{<t})$. For conditional generation, the probability is additionally conditioned on other inputs, which are omitted in our discussions without loss of generality.

We consider a natural language modeling task as the problem of modeling a finite set of pairs of a context and its conditional next-token distribution $\mathcal{L} = \{(c_1, P^*(X|c_1)), \cdots, (c_N, P^*(X|c_N))\}$, where $N$ is the number of possible contexts. The validity of the finiteness assumption has been discussed in [19] and does not affect our conclusion that follows.

A commonly-used approach for language modeling is to use neural networks to encode the context and the next token into vector representations $\mathbf{h}_c$ and $\mathbf{w}_x$ respectively. The conditional distribution is then modeled by a softmax function, $P_\theta(x|c) = \frac{\exp \mathbf{h}_c^\top \mathbf{w}_x}{\sum_{x'} \exp \mathbf{h}_c^\top \mathbf{w}_{x'}}$ where $\theta$ denotes the model parameters. The dot products between the two embeddings are called *logits*, and the corresponding feature space is termed a *logit space*.

We write down the context embeddings, token embeddings, and log probabilities in matrix forms as follows,

$$\mathbf{H}_\theta = \begin{bmatrix} \mathbf{h}_{c_1}^\top \\ \mathbf{h}_{c_2}^\top \\ \cdots \\ \mathbf{h}_{c_N}^\top \end{bmatrix}; \quad \mathbf{W}_\theta = \begin{bmatrix} \mathbf{w}_{x_1}^\top \\ \mathbf{w}_{x_2}^\top \\ \cdots \\ \mathbf{w}_{x_M}^\top \end{bmatrix}; \mathbf{A} = \begin{bmatrix} \log P^*(x_1|c_1) & \cdots & \log P^*(x_M|c_1) \\ \log P^*(x_1|c_2) & \cdots & \log P^*(x_M|c_2) \\ \vdots & \ddots & \vdots \\ \log P^*(x_1|c_N) & \cdots & \log P^*(x_M|c_N) \end{bmatrix}$$

where $M$ is the number of possible next tokens.

The language modeling problem is now turned into a matrix factorization problem of finding model parameters $\theta$ such that

$$\mathbf{H}_\theta \mathbf{W}_\theta^\top = \mathbf{A} + \text{row-wise shift} \tag{1}$$

The row-wise shift operation is defined as $\mathbf{A} + \mathbf{\Lambda}\mathbf{J}_{N,M}$ where $\mathbf{\Lambda}$ is a diagonal matrix with size $N \times N$ and $\mathbf{J}_{N,M}$ is an all-ones matrix with size $N \times M$.

Given the matrix factorization formulation, it follows that the rank of LHS in Eq. (1) is upper bounded by the embedding size $d$. Based on this key observation, the softmax bottleneck problem is identified as follows.

**Corollary 1** *(Softmax bottleneck) [19] If* $d < rank(\mathbf{A}) - 1$*, for any function family* $\mathcal{U}$ *and any model parameter* $\theta$*, there exists a context c in* $\mathcal{L}$ *such that* $P_\theta(X|c) \neq P^*(X|c)$*.*

In other words, given that most neural language models use distributed low-dimensional context and token embeddings, the softmax bottleneck indicates that these models do not have sufficient expressiveness to model complex, high-rank natural language.

## 3   Breaking the Softmax Bottleneck Efficiently

Mixture of Softmaxes (MoS) [19] is an effective approach to break the softmax bottleneck. Specifically, MoS uses the following formulation for the conditional distribution, $P_\theta(x|c) =$

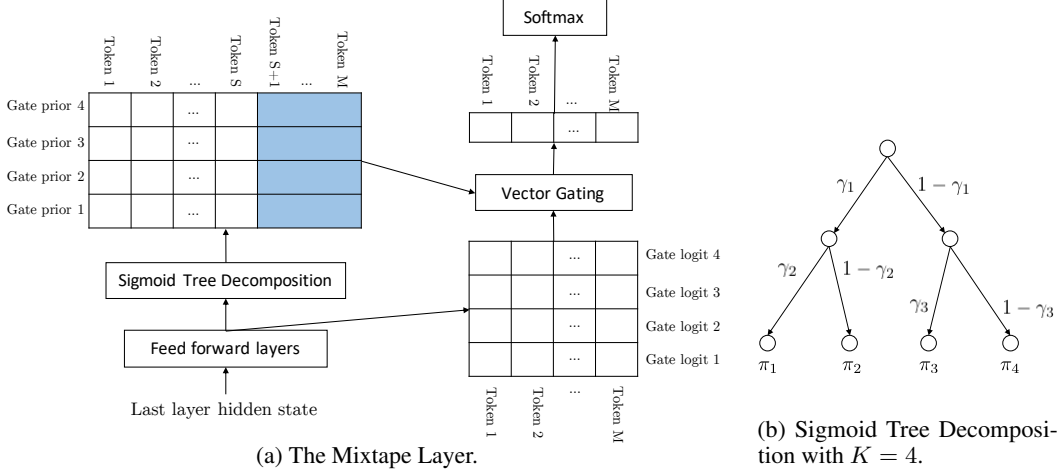

(a) The Mixtape Layer.

(b) Sigmoid Tree Decomposition with $K = 4$.

Figure 1: **Left**: $S$ is the number of *frequent tokens* that use their own gate priors, $M$ is the vocabulary size, and the blue boxes denote the gate priors shared by all *infrequent tokens*. In the diagram, the number of gates is set at $K = 4$, which is the value we use throughout the experiments. In our implementation, we do not explicitly compute the gate logits for infrequent tokens. Instead, we perform a scalar mixture using the shared gate priors and the context embeddings in Eq. (5) before multiplication with token embeddings to save memory. **Right**: Each edge in the sigmoid tree $\gamma_*$ is a probability computed using sigmoid functions. Each gate prior is the product of the probabilities along the path from the root to the leaf; e.g., $\pi_1 = \gamma_1 \gamma_2$.

$\sum_{k=1}^{K} \pi_{c,k} \frac{\exp \mathbf{h}_{c,k}^{\top} \mathbf{w}_x}{\sum_{x'} \exp \mathbf{h}_{c,k}^{\top} \mathbf{w}_{x'}}$; s.t. $\sum_{k=1}^{K} \pi_{c,k} = 1$ where the priors $\pi_{c,k}$ are obtained by another softmax-based function of the last-layer hidden states, and $K$ is the number of mixture components. This formulation is not limited by the softmax bottleneck because the log probability matrix $\mathbf{A}$ is modeled by $\hat{\mathbf{A}}_{\text{MoS}} = \log \sum_{k=1}^{K} \mathbf{\Pi}_k \exp(\mathbf{H}_{\theta,k} \mathbf{W}_{\theta}^{\top})$ where the log-sum-exp nonlinearity produces a high-rank matrix $\hat{\mathbf{A}}_{\text{MoS}}$.

However, softmax involves applying nonlinear `exp` transformations for each token in the vocabulary, performing reduction across the vocabulary, followed by division, which are all computationally intensive. Moreover, softmax is memory intensive because it has to store the pre-activations $\mathbf{h}_*^{\top} \mathbf{w}_*$, the post-activations $\exp(\cdot)$, and the output probabilities for each token in the vocabulary. Since a normal vocabulary size is in the magnitude of $10^4$, MoS dramatically increases the computational cost by using multiple softmaxes.

Another approach to break the softmax bottleneck was recently introduced [9]. However, this approach is less efficient than MoS because it computes a mixture of sigmoid functions in addition to softmaxes.

To alleviate the efficiency issue, we will introduce our novel method Mixtape that improves the efficiency over MoS without sacrificing the ability to learn high-rank representations.

## 3.1 Logit Space Vector Gating

Since the most expensive part of MoS is to compute $K$ softmaxes, significant computational budget can be saved if we manage to use only one softmax to compute the final probability distribution. It is tempting to move the mixture from the probability space into the logit space; i.e., mixing the representations before the softmax operation. This leads to the following conditional distribution,

$P_{\theta}(x|c) = \frac{\exp\left(\sum_{k=1}^{K} \pi_{c,k} \mathbf{h}_{c,k}\right)^{\top} \mathbf{w}_x}{\sum_{x'} \exp\left(\sum_{k=1}^{K} \pi_{c,k} \mathbf{h}_{c,k}\right)^{\top} \mathbf{w}_{x'}}$. However, as pointed out in [19], such a formulation will result in a low-rank representation because the matrix factorization form in Eq. (1) still applies.

Nevertheless, we will now show that with a small modification, applying mixture operations in the logit space leads to high-rank representations. The key idea is to use a *vector gating* mechanism instead of scalar mixtures. In other words, instead of using a shared set of mixture weights for every token, we use a different set of weights for different tokens. Formally, with vector gating, the

conditional distribution can be written as

$$P_\theta(x|c) = \frac{\exp \sum_{k=1}^{K} \pi_{c,x,k} \mathbf{h}_{c,k}^\top \mathbf{w}_x}{\sum_{x'} \exp \sum_{k=1}^{K} \pi_{c,x',k} \mathbf{h}_{c,k}^\top \mathbf{w}_{x'}}; \text{s.t.} \sum_{k=1}^{K} \pi_{c,x,k} = 1 \tag{2}$$

The log probability matrix $\mathbf{A}$ is now modeled as $\hat{\mathbf{A}}_{\text{Mixtape}} = \sum_{k=1}^{K} \mathbf{\Pi}_k \odot \left( \mathbf{H}_{\theta,k} \mathbf{W}_\theta^\top \right)$ Due to the elementwise multiplication introduced, the matrix factorization form in Eq. (1) does not apply, and the log probability matrix is therefore high-rank. In addition, the vector gating mechanism removes the necessity of computing $K$ softmax probability distributions, which makes efficiency improvement possible.

However, there is still a remaining obstacle before Mixtape is actually efficient enough. Notice that since the priors $\pi_{c,x,k}$ need to sum to one[1] for each context-token pair $(c, x)$, a naive implementation requires computing a softmax for the prior probabilities for each pair token $x$ given the context $c$. Let $l_{c,x,k}$ be the pre-activation priors, we have $\pi_{c,x,k} = \frac{\exp l_{c,x,k}}{\sum_{k'=1}^{K} \exp l_{c,x,k'}}$ Unfortunately, this will be even slower than MoS because the number of tokens in the vocabulary is usually large. In the following, we will introduce a novel technique that avoids such an efficiency trap.

## 3.2 Sigmoid Tree Decomposition

Now, we introduce how to efficiently compute the priors $\pi_{c,x,k}$. Instead of using a softmax, we propose to decompose a softmax distribution into a tree structure of sigmoid functions. Specifically, we compute $(K-1)$ sigmoid outputs and use them to define the probabilities along the tree branches. For example, with $K = 4$, the priors are defined as:

$$\begin{aligned}
\gamma_{c,x,k} &= \sigma(l_{c,x,k}) \text{ for } k = 1 \ldots K-1 \\
\pi_{c,x,1} &= \gamma_{c,x,1}\gamma_{c,x,2} \\
\pi_{c,x,2} &= \gamma_{c,x,1}(1 - \gamma_{c,x,2}) \\
\pi_{c,x,3} &= (1 - \gamma_{c,x,1})\gamma_{c,x,3} \\
\pi_{c,x,4} &= (1 - \gamma_{c,x,1})(1 - \gamma_{c,x,3})
\end{aligned} \tag{3}$$

where $\gamma_*$ denotes the sigmoid probabilities and $\sigma$ is the sigmoid function. The above equations are illustrated in Figure **??**.

We call this technique *sigmoid tree decomposition*. Such a decomposition is able to fully recover a $K$-way probability distribution with $(K-1)$ sigmoid functions. Using sigmoid functions removes the reduction and division operations in softmax and is more efficient.

Although the sigmoid tree composition technique can be used with any $K$, in our experiments, we always use $K = 4$ for two reasons. First, we find Mixtape is effective with $K = 4$ for all the tasks in our experiments. Second, speed is core to Mixtape and we fix $K$ to be the minimal possible value. Compared to MoS, using a fixed number of components $K$ means Mixtape requires less hyperparameter tuning efforts. Moreover, $K = 4$ is relatively small compared to the number of components in MoS, which further reduces the computational cost.

Let $\mathbf{g}_c$ be a $d_1$-dimensional last-layer hidden states given context $c$. The pre-activation priors $l_*$ are computed as

$$l_{c,x,k} = \mathbf{v}_x^\top \tanh(\mathbf{U}_k \mathbf{g}_c) + \mathbf{u}_k^\top \mathbf{g}_c + b_{x,k} \tag{4}$$

where $\mathbf{v}_x \in \mathbb{R}^{d_2}$, $\mathbf{U}_k \in \mathbb{R}^{d_2 \times d_1}$, $\mathbf{u}_k \in \mathbb{R}^{d_1}$, and $b_{x,k} \in \mathbb{R}$ are model parameters. Here $d_2$ is a hyperparameter that denotes the gate embedding size and is usually chosen to be much smaller than the normal word embedding size $d$. The context embeddings are obtained by

$$\mathbf{h}_{c,k} = \tanh(\mathbf{H}_k \mathbf{g}_c) \tag{5}$$

where $\mathbf{H}_k \in \mathbb{R}^{d \times d_1}$ is a model parameter.

### 3.3 Gate Sharing

So far we have arrived at an efficient high-rank model, but there is still room for further improvement. One observation is that we still have to compute a gate prior for each token in the vocabulary, which becomes the bottleneck of efficiency. However, for infrequent tokens, it is hard to estimate the gate priors accurately due to lack of training samples, and thus learning gate priors for infrequent tokens might simply be waste of computation. As a way to leverage this observation, the core idea of *gate sharing* is to share the same gate priors for all infrequent words. Specifically, for an infrequent token $x$, the pre-activation gate priors are defined as

$$l_{c,x,k} = \mathbf{u}_k^\top \mathbf{g}_c \tag{6}$$

which remains constant given $c$ and $k$ for different infrequent tokens $x$.

The resulting representations are partially high-rank. Supposed the token indices are ranked by frequency. The log probability matrix is now modeled by

$$\hat{\mathbf{A}} = \left[ \sum_{k=1}^K \mathbf{\Pi}_k^{(1)} \odot \left( \mathbf{H}_{\theta,k}^{(1)} \mathbf{W}_\theta^{(1)\top} \right) ; \mathbf{H}_\theta^{(2)} \mathbf{W}_\theta^{(2)} \right]$$

where the superscripts $(1)$ and $(2)$ denote the representations for frequent and infrequent tokens respectively. We have high-rank and low-rank representations for frequent and infrequent tokens respectively. For infrequent tokens, our formulation is equivalent to performing logit space scalar mixtures, also known as Mixture of Contexts in [19]. Similar ideas have been demonstrated in previous work [8] where infrequent tokens use less-expressive representations (smaller embedding sizes) to save memory and computation without affecting performance.

With gate sharing, we use the shared gate prior to mix the context embedding $\mathbf{h}_{c,k}$ before multiplication with the token embeddings $\mathbf{w}_x$, which saves memory because no gate logits are stored for infrequent tokens. Gate sharing also speeds up the computation by computing only one set of gate priors for all infrequent tokens.

Let $S$ be the number of frequent tokens and let $r = S/M$ with $M$ being the vocabulary size. In our experiments, we set $r = 0.5$ for machine translation and $r = 0.1$ for language modeling.

### 3.4 Summary and Discussion

The Mixtape layer is summarized as follows:

1. Given the last-layer hidden states $\mathbf{g}_c$, compute the context embeddings $\mathbf{h}_{c,k}$ using Eq. (5).

2. For each frequent token $x$, compute the pre-activation gate priors $l_{c,x,k}$ using Eq. (4).

3. For all infrequent tokens, compute a shared pre-activation gate prior $l_{c,x,k}$ using Eq. (6).

4. Use sigmoid tree decomposition to compute the gate priors $\pi_{c,x,k}$ as in Eq. (3).

5. Use vector gating to obtain the next-token probabilities using Eq. (2).

The architecture of the Mixtape layer is illustrated in Figure 1.

In our implementation, we also add biases to the matrix multiplication operations in Eq. (2), (4) and (5), which were omitted in the above text for simplicity. It is also optional to employ weight normalization [14] for the parameter $\mathbf{U}_k$ in Eq. (4). Different from [14], we use a constant scale instead of a learnable one as it leads to more stable optimization. In our experiments, we use weight normalization for language modeling but did not observe improvement on machine translation tasks. We also apply dropout on $\tanh(\mathbf{U}_k \mathbf{g}_c)$ and $\mathbf{h}_{c,k}$ in Eq. (4) and (5). To further regularize the networks, we also add a small amount of Gaussian noise on the pre-activation priors $l_*$ in the forward pass.

If we neglect cheap operations and only consider matrix multiplication and softmax, MoS has $2(d_1 dK + dKM)$ FLOPs for matrix multiplication and $K$ $M$-way softmaxes. For comparison, Mixtape has $2(d_1 dK + dKS)$ FLOPs for matrix multiplication and one $M$-way softmax. The speedup of Mixtape comes from a smaller number of softmaxes, a smaller $K$, and a smaller $S < M$. Suppose an $M$-way softmax uses $8M$ bytes for storing intermediate and final results. If we again only consider major operations of matrix multiplication and softmax, with FP32 tensors, MoS roughly uses $(4dK + 12MK)$ bytes and Mixtape uses $(4dK + 12SK + 8M)$ bytes. Mixtape uses less memory due to a smaller $S$ and a smaller $K$.

| Method | En-De BLEU | En-Fr BLEU |
|---|---|---|
| [18] – Transformer | 28.4 | 41.0 |
| [6] – Universal Transformer | 28.9 | - |
| [1] – Weighted Transformer | 28.9 | 41.4 |
| [16] – Transformer + Relative encodings | 29.2 | 41.5 |
| [10] – Transformer + MoS | 29.5 | 42.1 |
| [13] – Large-batch training | 29.3 | 43.2 |
| [17] – Mesh Tensorflow (2.9B params) | 26.7 | 43.9 |
| [17] – Mesh Tensorflow (0.8B params) | 27.5 | 43.5 |
| Ours – Transformer + Mixtape (0.2B/0.8B params) | 29.3 | 43.9 |

Table 1: Comparison with state-of-the-art systems on WMT En-De and En-Fr. Mixtape uses 0.2 and 0.8 billion parameters for En-De and En-Fr tasks respectively.

| Dataset | Size | Unit | Vocab Size | Dataset | Size | Unit | Vocab Size |
|---|---|---|---|---|---|---|---|
| PTB | 1M tokens | Word | 10K | En-De | 4.5M pairs | BPE | 32K |
| 1B | 1B tokens | Word | 100K | En-Fr | 36M pairs | BPE | 32K |

Table 2: Dataset statistics. "PTB" and "1B" denote Penn Treebank and One Billion Word respectively.

# 4 Experiments

Our experiments consist of three parts. First, we demonstrate that the proposed Mixtape layer is able to improve state-of-the-art machine translation systems by breaking the softmax bottleneck. Second, we compare the perplexity, translation quality, speed, and memory constraints of Mixtape, MoS, and softmax, to demonstrate that Mixtape is able to achieve a good balance between effectiveness and efficiency. Third, through ablation studies, we show the benefits of gate sharing.

## 4.1 Datasets

We test Mixtape on two tasks, language modeling and machine translation. For language modeling, we exactly follow the settings in [19] on Penn Treebank [12] and One Billion Word [4] for fair comparison. We implement the same recurrent network architectures and follow the regularization and optimization techniques used in [19]. We tune the model size of Mixtape such that Mixtape has the same number of parameters as MoS in the corresponding settings. On One Billion Word, we also replicate the data preprocessing pipeline that lower-cases the text and chooses the top 100K tokens as the vocabulary. This results in a non-standard setting, but it enables fair comparison with MoS as well as excluding the orthogonal effects of techniques for a larger vocabulary such as adaptive softmax [8].

For machine translation, our experiments are based on two widely-used WMT'14 benchmarks, English to German (En-De) and English to French (En-Fr), following the setups in [13, 18]. For En-De, we train on the WMT'16 training data and test on `newstest14`. For En-Fr, we train on the WMT'14 training data and test on `newstest14`. We use BPE encodings [15] with a vocabulary size of 32K. Following [17], we use `sacrebleu` for evaluation.

The statistics of different datasets and settings are shown in Table 2. The selected datasets present a degree of diversity in sizes, input units, and vocabulary sizes, which enables evaluating the robustness of Mixtape.

## 4.2 WMT'14 Results

We apply Mixtape on top of Transformers [18] to have a comparison with state-of-the-art systems on WMT'14 benchmarks. We also incorporate relative positional encodings [16] in our architecture. On En-De, we employ a 6-layer Transformer with embedding size 1024, inner layer size 4096, and 16 attention heads. We train for 300K steps with a learning rate of 2.5, a batch size of 4096, and 16K warmup steps. We apply a dropout of 0.3 on the layer outputs, a dropout of 0.15 on attention probabilities, a dropout of 0.2 on $\tanh(\mathbf{U}_k\mathbf{g}_c)$ in Eq. (4), and a Gaussian noise with 0.1 stdev on pre-activation gate priors. On En-Fr, we employ a 6-layer Transformer with embedding size 2048, inner layer size 8192, and 16 attention heads. We train for 1.2M steps with a learning rate of 2.0, a batch size of 4096, and 16K warmup steps. We apply a dropout 0.25 on the layer outputs, dropouts of

| Method | Perplexity | | Perplexity | | Training time (same bsz) | | Training time (same mem) |
|---|---|---|---|---|---|---|---|
| | Prior work | Our impl. | Out layer | All layers | Out layer | All layers |
| Softmax | 58.8 | 59.19 | 14 | 328 | 0.40 | 3.06 |
| MoS-3 | 58.62 | 57.62 | 147 | 439 | 3.37 | 6.25 |
| MoS-5 | 57.36 | 57.24 | 173 | 488 | 4.40 | 8.38 |
| MoS-10 | 56.33 | 56.49 | 242 | 609 | 6.96 | 12.99 |
| MoS-15 | 55.97 | **56.14** | 310 | 731 | 9.62 | 15.73 |
| Mixtape | - | 56.37 | 27 | 345 | 0.90 | 3.66 |
| - no sharing | - | *56.33* | 95 | 487 | 3.88 | 10.44 |

Table 3: Perplexity and training time comparison on Penn Treebank. "MoS-$K$" means MoS with $K$ mixture components, "no sharing" means using Mixtape without the gate sharing technique, "our impl." means results from our own implementation, "out layer" means training time for the output layer only, "all layers" means training time for the entire network, "same bsz" means using the same batch size of 48, "same mem" means using the same GPU memory budget of 12GB with a maximum possible batch size. Results from prior work are taken from [19]. In the setting with a fixed batch size ("same bsz"), training time per batch in seconds is reported. In the setting with a fixed memory budget ("same mem"), training time per instance in seconds is reported.

| Method | Perplexity | Training time (same bsz) | | Training time (same mem) | |
|---|---|---|---|---|---|
| | | Out layer | All layers | Out layer | All layers |
| Softmax | 42.77 | 53 | 119 | 3.2 | 7.0 |
| MoS-7 | 37.10 | 794 | 856 | 52.5 | 59.8 |
| Mixtape | **36.52** | 114 | 170 | 8.6 | 11.3 |
| - no sharing | 36.77 | 364 | 414 | 34.3 | 48.6 |

Table 4: Perplexity and training time comparison on One Billion Word. Text abbreviations are the same as Table 3. In the setting with a fixed batch size ("same bsz"), we use a batch size of 20. Results of Softmax and MoS-7 are taken from [19].

0.15 on attention probabilities and $\tanh(\mathbf{U}_k \mathbf{g}_c)$ in Eq. (4), and a Gaussian noise with 0.1 stdev on pre-activation gate priors.

The results of our method are shown in Table 1. Mixtape with Transformers achieves state-of-the-art results on both En-De and En-Fr. Interestingly, Mixtape outperforms baselines that use MoS [10]. This demonstrates that breaking the softmax bottleneck significantly contributes to achieving state-of-the-art performance for machine translation, and Mixtape is an effective approach to break such a bottleneck. On En-Fr, Mixtape obtains the same performance with Transformers trained with Mesh Tensorflow [17]. However, Mixtape is much more parameter-efficient, using only 0.8 billion parameters v.s. 2.9 billion parameters in Mesh Tensorflow. Moreover, Mixtape outperforms Mesh Tensorflow by a large margin on En-De, demonstrating more robustness and generalization capabilities on relatively small datasets. Note that [7] reports better performance with back translation, which is not comparable with our setting.

## 4.3 Ablation Study and Comparison with Baselines

We now compare the performance of Mixtape with MoS and softmax, as well as studying the effects of gate sharing. We report the training time used for both the sole output layer and the entire network. To take the memory usage of different methods into consideration, in addition to reporting training time with the same batch size, we also consider the training time with the same memory budget. In other words, a model that uses more memory will have a smaller batch size, and thus training time per instance will increase.

The results of different methods on Penn Treebank, One Billion Word, WMT'14 En-De, and WMT'14 En-Fr are shown in Tables 3, 4, 5, and 6. We use baseline MoS results from [19, 10] whenever possible and avoid using our own implementation for fair comparison. There are three main messages delivered in these results.

First, compared to softmax, Mixtape is comparably efficient while being more accurate at language modeling and translation. On tasks with normal vocabulary sizes including Penn Treebank, WMT'14 En-De, and WMT'14 En-Fr, a Mixtape-based network is only 5% to 18% slower than a softmax-based network given the same batch size and only 20% to 34% slower given the same memory budget. Even on One Billion Word with a 100K vocabulary, a Mixtape-based network is only 60% slower

| Method | BLEU | Training time (same bsz) | | Training time (same mem) | |
|---|---|---|---|---|---|
| | | Out layer | All layers | Out layer | All layers |
| Softmax | 29.0 | 2.16 | 18.15 | 5.4 | 37.0 |
| MoS-9 | 29.5* | 14.36 | 30.08 | 61.1 | 97.9 |
| Mixtape | 29.3 | 5.83 | 21.48 | 17.6 | 49.8 |

Table 5: BLEU and training time comparison on WMT'14 En-De. Text abbreviations are the same as in Table 3. * indicates results taken from [10]. In the setting with a fixed batch size ("same bsz"), we use a batch size 256 and report the training time per 100 batches in seconds. In the setting with a fixed memory budget ("same mem"), training time per instance in milliseconds is reported.

| Method | BLEU | Training time (same bsz) | | Training time (same mem) | |
|---|---|---|---|---|---|
| | | Out layer | All layers | Out layer | All layers |
| Softmax | 43.0 | 2.43 | 24.06 | 15.0 | 159.3 |
| MoS-9 | 42.1* | 7.90 | 29.05 | 254.0 | 936.0 |
| Mixtape | 43.9 | 4.88 | 26.81 | 40.5 | 197.2 |

Table 6: BLEU and training time comparison on WMT'14 En-Fr. Text abbreviations are the same as in Table 3. The way we report the training time for different settings is the same as in Table 5.

than a softmax-based network. On the other hand, Mixtape improves the perplexity over MoS by 2.8 points and 6.25 points on Penn Treebank and One Billion Word respectively. On translation tasks, Mixtape improves the BLUE scores from 29.0 to 29.3 on En-De and from 43.0 to 43.9 on En-Fr.

Second, compared to MoS, Mixtape achieves similar or better performance in perplexity and BLEU while being much more efficient. Mixtape is 1.6x to 11.5x faster than MoS given the same batch size and 3.5x to 10.5x faster given the same memory budget. The speedup is usually more significant with the memory budget constraints, demonstrating that the ability to save memory also contributes to the efficiency of Mixtape. Mixtape has better performance than MoS on translation and comparable performance on language modeling.

Third, gate sharing substantially reduces the computational cost without sacrificing accuracy. In Tables 3 and 4, the perplexities of Mixtape with and without gate sharing only have an almost negligible difference. Gating sharing improves the speed by 4.3x and 4.0x on Penn Treebank and One Billion Word respectively given the same memory budget. The speedup is 3.5x and 3.2x given the same batch size. This indicates that gate sharing reduces the memory cost as well as training time per forward-backward pass.

## 5  Conclusions

We propose Mixtape to break the softmax bottleneck more efficiently. Compared to MoS, Mixtape is more computationally efficient. Compared to softmax, Mixtape has comparable efficiency and is superior in terms of accuracy. Based on the above results, it is possible that Mixtape can be used as a plug-and-play layer to improve conditional and unconditional text generation in general. In the future, it will be intriguing to further investigate more applications of Mixtape.

## Footnotes

[1]We were not able to get good performance with unnormalized priors.

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
