[Reviews · NeurIPS 2019]

Reviewer 1



This work deals with the softmax bottleneck problem, which is noticed by Yang et al. (2017). Departing from the mixture of softmax used by previous work, which is computationally expensive, this work proposes to elementwise gating technique, which could potentially get around the softmax bottleneck problem by taking one single softmax computation. The gating vectors are efficiently computed with a series of sigmoid functions organize in a tree structure, and gate sharing is used to further promote efficiency. Empirical evaluation on machine translation and language modeling shows that the proposed method is able to achieve comparable accuracy to the mixture of softmax baseline with much less computation cost. Overall I think this is a solid work: it clearly presents an interesting idea, which yields strong empirical performance. Details: - Broken pointer in line 132. - Line 115, on the rank of \hat{A}_{Mistape}. If I understand it correctly, $\Pi_k$ matrix and elementwise product could bump the rank up, due to rank(A\odot B) \leq rank(A)rank(B). Therefore a Mixtape model with K=1 could potentially yield a high-rank log probability matrix. Have the authors considered comparing to this baseline? - Adding onto the above point, I suggest the authors tone down a bit the argument that it is `therefore high-rank`, unless a lower bound can be derived (which I think is tricky to do). And it would be interesting to empirically see its rank in practice.

Reviewer 2



POS-AUTHOR FEEDBACK I thank the authors for their feedback and clarifications. I have increased my score based on those answers, and trusting that the promised modifications will appear in the final version. I would strongly encourage to make the release of the code as easy to use as possible, ideally with plugins for major platforms. This would not only increase citations, but have a direct impact in a number of use-cases ORIGINAL REVIEW This paper addresses the softmax bottleneck problem: resolving it has shown to significantly improve results when the output is over a large space (eg: NLP). However, current solutions are very costly. This papers contributes with a tradeoff between efficiency and cost: it obtains worse results than the full mixture-of-softmax, but does so much cheaper. There is much to like of this paper, as it contributes an important tool. I believe however that the impact would be much higher if the authors would provide a “plug-and-play” layer for at least one popular deep learning toolkit. People don’t use the best algorithm, the use the best one available. Improvement-wise: - Some technical comments are only glossed over and would merit a more detailed discussion: o Line 117: “is therefore high-rank”. This seems very important, but this comment is never proved or expanded upon o Same applies for line 157: “partially high-rank”. What does this mean? o Line 121 (footnote 1 p4): the priors need to sum to one. I don’t see why they need to. The footnote just details that worse performance are obtained. This seems rather a crucial point as solving it would render 3.2 and 3.3 unnecessary. Similarly, the author seems to assume that softmax is the only normalization technique. Why not trying simple (l1) norm? This would avoid the exp computation - There is a huge amount of hyper-parameter optimization going on. Different from what is said in the reproducibility criteria (“The range of hyper-parameters considered, method to select the best hyper-parameter configuration, and specification of all hyper-parameters used to generate results.”), it is never specified how this is done. This includes setting r (line 169), and non-standard decisions like adding Gaussian noise (185). At the same time, it is not clear what experiments were run by the authors: it seems the translation experiments were not, but then training time is reported in Table 5 - No comparison with [9] is reported Other comments: - it seems that hierarchical softmax could be a way of solving the efficiency problem of the softmax for MoS. As it shares the tree-structure idea of sigmoid tree decomposition, I believe this merits a discussion. - the gate sharing idea is reminiscent of some interpolation techniques from the time of n-gram LM (given different weight to frequent and unfrequent tokens). As at that time, this idea can be used at very different levels to bin parameters: not one per word or one for all unfrequent words but clustering them and sharing the gates across clusters. - Line 132: the Fig cross-reference is not resolved

Reviewer 3



- The motivation of the paper is clear and its contributions are clear (see section 1). - The method was well written. - Experiments are strong. - The paper is overall well written, but the preliminary parts explaining 'softmax bottleneck' and 'mixture of softmax' was hard to understand without knowing the original papers. - Need more ablation studies about the gate sharing ratio (r) and the number of sigmoids of the sigmoid tree (K) to provide further information about the proposed method to other researchers. - (Misc) Why the method name is Mixtape? Line 132: missing latex reference (??) -------------------------- The rebuttal addressed my concerns on their rebuttal. I believe the author will provide more detailed analysis and experiments in the final paper.

[Author Response · NeurIPS 2019]

We thank all reviewers for the valuable feedback. We will fix the formatting issues and address the concerns pointed out by the reviewers in the final manuscript.

**Response to Reviewer 1**:

**High rank with $K = 1$**: In our preliminary experiments, using $K = 1$ resulted in degraded performance. We believe this is because (1) $K = 1$ reduces the empirical rank and (2) using a large $K$ imposes a "branching" inductive bias that benefits training. We will compare the performance and empirical rank of different $K$'s in our revision.

**High-rank argument**: We believe this is a very good point. We will tone down about the high-rank argument that elementwise multiplication leads to high-rank representations because the argument is mainly empirical, though somewhat intuitive. We will also add a study about the empirical rank in our revision.

**Response to Reviewer 2**:

**Line 117**: The argument is mainly empirical. Intuitively, it is likely that the results of elementwise multiplication are high-rank because the features are randomly distributed. We will clarify this and add a study about the empirical rank in our revision.

**Line 157**: "partially high-rank" means that frequent tokens have high-rank representations and infrequent tokens have low-rank representations (lines 159-161). We will further clarify this.

**Line 121**: It is an open question why unnormalized priors do not work. We conjecture that there might be two reasons: (1) normalization introduces competition among different branches, which encourages different branches to learn different features; (2) normalized inputs have more stable norms, which might lead to more stable optimization. This is similar to the attention mechanism where normalizing the attention features with probabilities summing to one is necessary for the best performance. We believe these are very important points. However, as understanding and improving prior normalization (e.g. using L1 normalization) is largely nontrivial, we leave them to future work.

**Hyperparameters**: For the value $r$, we try $r = 0.1$ and $r = 0.5$. The performance of $r = 0.5$ is slightly better than or equal to $r = 0.1$, but we found the gains of using $r = 0.5$ are small for LM in preliminary experiments so we use $r = 0.1$ for LM. We fix the Gaussian noise at $0.1$ for all experiments. The other hyperparameters are shared by MoS and Mixtape and we use the same hyperparameter search space for the two methods. We performed random search and ensure the same number of trials are used for the two methods. The search space includes: dropout [0.0, 0.1, 0.3], learning rate [0.1, 0.2]. We will include the numbers and clarify the settings in our revised paper.

**Comparison with [9]**: We mainly focus on improving the efficiency in this paper. Because [9] is more expensive than MoS, we use MoS as our direct baseline. We will include [9] in our tables to give readers more information.

**Other possibilities**: We believe that the ideas of hierarchical softmax and word clustering are appealing, which are interesting directions for future work. We will also include the related work in our updated version.

**Code**: We will publish our code for reproducing all of our results in this paper.

**Response to Reviewer 3**:

**Background**: We will add more details to the background section, better explaining 'softmax bottleneck' and 'mixture of softmaxes'.

**Hyperparameters and ablation study**: We believe it is valuable to perform a study to understand the effects of different values of $K$ and $r$. In our experiments, we fix $K = 4$ throughout all experiments to minimize the effects of hyperparameter tuning, and this value is recommended for future use of our method. We experimented with $r = 0.1$ and $r = 0.5$ in our early experiments, and found the two values have similar performance on language modeling, while using $r = 0.5$ yields improvement of about 0.1 to 0.3 BLEU for machine translation. As a result, we believe our model is not very sensitive to these hyperparameters and the default values will be sufficient for most tasks. We will provide more detailed analysis and comparisons in our final version of the paper.

**Why Mixtape**: We use this name because the "mix" prefix is related to our approach of mixing the logits.

[Meta-Review · NeurIPS 2019]

This paper proposes techniques to deal with the softmax bottleneck problem. Pros • Experimental results show strong performances in language modeling and machine translation. • The paper is clearly written. • The proposed techniques are technically sound and novel. Cons • Writing of the paper can be further enhanced by making it self-contained. • More experiments can be conducted. The paper represents solid work. There are clarity issues pointed out by the reviewers. The authors have promised to provide more details and experiments in their final version. They also promise to release the codes.